# Association of Metabolic Syndrome with the Risk of Head and Neck Cancer: A 10-Year Follow-Up Study of 10 Million Initially Healthy Individuals

**DOI:** 10.3390/cancers15164118

**Published:** 2023-08-15

**Authors:** Geun-Jeon Kim, Kyung-Do Han, Young-Hoon Joo

**Affiliations:** 1Department of Otolaryngology-Head and Neck Surgery, College of Medicine, The Catholic University of Korea, Seoul 06591, Republic of Korea; emelenciana@catholic.ac.kr; 2Department of Biostatistics, College of Medicine, The Catholic University of Korea, Seoul 06591, Republic of Korea; hkd@ssu.ac.kr

**Keywords:** head and neck neoplasms, metabolic syndrome, smoking, epidemiology, Republic of Korea

## Abstract

**Simple Summary:**

The global incidence of metabolic syndrome is on the rise, and its association with various types of cancer has been established. However, research into an association between metabolic syndrome and the incidence of head and neck cancer has been relatively limited. This population-based study aimed to report on the association of metabolic syndrome with head and neck cancer and individual subtypes of cancers. The study found that participants with metabolic syndrome exhibited a higher risk of developing head and neck cancer, particularly oral cavity and laryngeal cancers. The elevated levels of fasting glucose and blood pressure present in individuals with metabolic syndrome were independently associated with an increased risk of head and neck cancer. The results of this study could assist with etiological investigations and prevention strategies.

**Abstract:**

The aim of this national population-based retrospective study was to analyze the relationship between MetS and the incidence of HNC. In this Korean population-based cohort study, 9,598,085 subjects above the age of 20 were monitored from 1 January 2009 to 31 December 2018. In the study population, a total of 10,732 individuals were newly diagnosed with HNC during the 10-year follow-up. The hazard ratio (HR), after adjusting for age, gender, smoking status, alcohol intake, and exercise, indicated that participants with MetS were at a 1.06-fold (95% CI: 1.01–1.10) higher risk of having HNC than those without MetS. Participants with MetS showed an increased risk of developing oral cavity cancer (HR, 1.12; 95% CI, 1.03–1.23) and laryngeal cancer (HR, 1.18; 95% CI, 1.09–1.27). Among the components of MetS, elevated fasting glucose (HR = 1.04, 95% CI: 1.00–1.08) and elevated blood pressure (HR = 1.08, 95% CI: 1.04–1.13) were significantly associated with an increased HR for HNC in an adjusted multivariable model. The association between HNC and MetS remained significant even among individuals who had never smoked or were ex-smokers (HR: 1.09; 95% CI: 1.04–1.15), as well as those who did not drink or were mild drinkers (HR: 1.07; 95% CI: 1.02–1.12). The findings of this cohort study suggest MetS was associated with an increased risk for some types of HNCs. The results of this study could assist with etiological investigations and prevention strategies.

## 1. Introduction

Metabolic syndrome (MetS) encompasses a combination of metabolic abnormalities such as insulin resistance, obesity, hypertension, hyperglycemia, dyslipidemia, and elevated triglyceride levels [1]. The global incidence of MetS is persistently increasing, with a notable surge in developed countries where diabetes and obesity rates are on the rise [2,3]. Head and neck cancer (HNC) was the seventh most common cancer in 2020 with 931,931 new cases globally [4], and includes oral cavity, nasopharyngeal, oropharyngeal, hypopharyngeal, laryngeal, and salivary gland cancers. In Korea, the incident cases of HNC have been steadily increasing over the past decade from around 4100 individuals in 2010 to 5600 individuals in 2020.

Smoking and alcohol consumption are major risk factors for HNC [5], but the relationship between obesity and the incidence of HNC remains controversial [6,7,8,9]. In a comprehensive analysis involving more than six million men from 203 studies, Petrelli et al. [10] found evidence indicating a higher overall cancer mortality rate among obese men. Additionally, adults classified as having metabolically healthy obesity displayed an elevated susceptibility to developing cancer [11].

MetS is associated with numerous diseases. Recent studies have demonstrated an increased risk of several cancers, including liver, colorectal, bladder, endometrial, pancreatic, and breast cancers [12,13,14,15]. Furthermore, meta-analyses indicated a strong association between MetS and thyroid cancer [16] and thyroid nodules [17]. Notably, the prevalence of MetS is progressively rising among Korean adults, emphasizing the need for population-based investigations to validate the relationship between MetS and cancer occurrence [18]. However, the association between the incidence of HNC and MetS has not been clearly determined. Thus, the objective of this study was to investigate the association between MetS and the risk of HNC and its subtypes.

## 2. Materials and Methods

### 2.1. Source of Data

Administered by the Ministry of Health, Welfare, and Family Affairs [19], the Korean National Health Insurance Service (KNHIS) functions as a public medical insurance system [19]. Operating as a mandatory social insurance program, the KNHIS initiative provides coverage for approximately 97% of the entire Korean population, while the remaining 3% are covered by the Medical Aid program [20]. The KNHIS database includes patient demographics and records of diagnoses, interventions, and prescriptions [21]. Diagnoses were confirmed using International Classification of Disease, Tenth Revision, Clinical Modification (ICD-10-CM) codes, which include C02, C03, C04, C05, and C06 for oral cavity cancer; C07 and C08 for salivary gland cancer; C11 for nasopharyngeal cancer; C01, C051, C099, and C103 for oropharyngeal cancer; C12 and C13 for hypopharyngeal cancer; C10 for sinonasal cancer; and C32 for laryngeal cancer. The study’s protocol was granted approval by the Institutional Review Board of the Catholic University of Korea in Seoul.

### 2.2. Patient Selection

Individuals covered by the National Health Insurance Corporation are advised to undergo standardized medical assessments every two years. Those who were over 20 years old and had undergone health checkups in 2009 were enrolled in this study (n = 10,585,852). We monitored subjects until 31 December 2018. We excluded individuals with missing data (n = 746,403) and those with a history of other cancers before the health checkup (n = 153,456). We also applied a one-year lag period to minimize detection bias (n = 87,908). The remaining 9,598,085 subjects (5,220,801 males and 4,377,284 females) were included in this study from baseline to the date of diagnosis of HNC. Participants were defined as having HNC if their National Health Insurance records from 2010 to 2018 documented admissions related to HNC. The medical assessments encompassed measurements of height, weight, and blood pressure. Furthermore, measurements of fasting plasma glucose, triglyceride, total cholesterol, and high-density lipoprotein (HDL) levels were acquired. Past medical history and information on health-related behaviors such as smoking, alcohol intake, and physical activity were collected using a standardized, self-reported International Physical Activity Questionnaire. Also, the household income classification of this study was based on health insurance premiums; the National Health Insurance beneficiaries were divided into quartiles.

### 2.3. Definition of Metabolic Syndrome

The assessment period for MetS was determined based on the health checkup conducted in 2009. The definition of MetS was based on that of the joint interim statement of the International Diabetes Federation (IDF) Task Force on Epidemiology and Prevention [9]. According to this institution, MetS comprises three or more of these five components: abdominal obesity based on population- or country-specific definitions (waist circumference ≥ 85 cm) [22], elevated blood pressure (systolic ≥ 130 and/or diastolic ≥ 85 mmHg), hyperglycemia (fasting plasma glucose ≥ 100 mg/dL), hypertriglyceridemia (triglycerides ≥ 150 mg/dL), and low HDL-cholesterol levels (<40 mg/dL). Body mass index (BMI) was calculated as weight in kilograms divided by the square of height in meters (kg/m^2^).

### 2.4. Statistical Analysis

Basic characteristics are presented using descriptive analysis. Differences in baseline characteristics between groups were determined using Student’s *t*-test for continuous variables and the χ^2^ test for categorical variables. HNC incidence was calculated by dividing the number of cases by 1000 person-years. Cox proportional hazards models were applied to estimate the hazard ratio (HR) and 95% confidence intervals (CIs) for the associations between MetS and the risk of HNC. Two linear regression models were constructed to explain the nonlinear relationship. First, segmented linear regression models were generated. The breakpoints were chosen as the values minimizing the Akaike information criteria of the Cox models. Second, multivariable regression analyses were performed using the number of metabolic components. Subtype analyses were also performed by using multivariable Cox proportional hazards models. Model ^1^ was unadjusted. Model ^2^ was adjusted for age, gender, smoking, alcohol consumption, regular exercise, and income. Model ^3^ was adjusted for age, gender, smoking, alcohol consumption, regular exercise, income, diabetes, and hypertension. Statistical analyses were performed using SAS version 9.2 (SAS Institute, Cary, NC, USA).

## 3. Results

### 3.1. Basic Characteristics

The characteristics of the study population are indicated in Table 1. During the 9-year follow-up period, 10,732 subjects were diagnosed with HNC. HNC patients were significantly older than those without HNC (*p* < 0.001). At baseline, the percentages of male and female participants with HNC were 79.2% and 20.8%, respectively (*p* < 0.001). Associations with current smoking, heavy alcohol intake, diabetes, hypertension, and dyslipidemia were statistically significant in the HNC group. The study revealed that HNC patients exhibited a substantially higher average waist circumference and higher systolic and diastolic blood pressure, fasting glucose, HDL cholesterol, and triglyceride levels compared to those without HNC. The number of MetS components was a risk factor with a higher risk estimate of HNC (*p* < 0.001).

### 3.2. Associations between Metabolic Syndrome and Head and Neck Cancer

The unadjusted and multivariable-adjusted HRs for having HNC according to the presence or absence of MetS are presented in Table 2. Among the data, an adjustment was performed based on factors and diseases that affected HNC. An HR adjusted for age, gender, smoking, alcohol consumption, regular exercise, income, diabetes, and hypertension indicated that participants with MetS had a 1.06-fold higher risk of having HNC than those without MetS (95% CI: 1.01–1.10). Participants with MetS showed an increased risk of developing oral cavity cancer (HR = 1.12, 95% CI: 1.03–1.23) and laryngeal cancer (HR = 1.18, 95% CI: 1.09–1.27). No significant association was observed between MetS and an incidence of salivary gland cancer, nasopharyngeal cancer, oropharyngeal cancer, hypopharyngeal cancer, and sinonasal cancer. The unadjusted and multivariable-adjusted HRs for having HNC according to the number of MetS components are presented in Appendix A. Figure 1 demonstrates the adjusted HRs (Model ^3^) from Appendix A, where the incidence of overall HNC shows a significant linear trend according to the score of metabolic syndrome components (*p* = 0.017). In the results by subtype, the incidence rates show a linear trend according to the score of metabolic syndrome components. However, statistically significant results were observed in the case of laryngeal cancer (*p* < 0.001) and oropharyngeal cancer (*p* = 0.032) (Appendix A).

### 3.3. Association between Individual Components of Metabolic Syndrome and Head and Neck Cancer

We performed further analyses according to the individual components of MetS (Table 3). The adjusted multivariable model shows that an increased risk of HNC was significantly associated with higher fasting glucose (HR = 1.04, 95% CI: 1.00–1.08) and elevated blood pressure (HR = 1.08, 95% CI: 1.04–1.13).

The results regarding the association between the individual components of MetS and HNC subtypes are presented in Appendix A. Subtype analysis revealed that a high waist circumference was associated with sinonasal cancer (HR = 1.23, 95% CI: 1.02–1.48), nasopharyngeal cancer (HR = 1.18, 95% CI: 1.03–1.35), and salivary gland cancer (HR = 1.17, 95% CI: 1.03–1.33). Conversely, a low waist circumference was associated with hypopharyngeal cancer (HR = 0.64, 95% CI: 0.55–0.76). Elevated fasting glucose was correlated with laryngeal cancer (HR = 1.13, 95% CI: 1.05–1.21). Elevated blood pressure was associated with laryngeal cancer (HR = 1.21, 95% CI: 1.12–1.31), hypopharyngeal cancer (HR = 1.27, 95% CI: 1.10–1.47), and oral cancer (HR = 1.12, 95% CI: 1.02–1.22). An elevation in triglyceride levels was associated with laryngeal cancer (HR = 1.22, 95% CI: 1.05–1.21) and oral cancer (HR = 1.12, 95% CI: 1.03–1.22). A low HDL cholesterol level was correlated with an increased risk of oral cancer (HR = 1.14, 95% CI: 1.04–1.24).

### 3.4. Associations between Other Risk Factors and Head and Neck Cancer

We examined the joint effect of MetS and age, gender, smoking status, alcohol consumption, exercise frequency, and BMI on the risk of HNC (Table 4). The risk of HNC was increased due to MetS regardless of gender (HR: 1.08, 95% CI: 1.03–1.13 for men; HR: 1.17, 95% CI: 1.06–1.28). HNC was associated with MetS even when individuals were never smokers or ex-smokers (HR: 1.09; 95% CI: 1.04–1.15) and had no or mild alcohol consumption (HR: 1.07; 95% CI: 1.02–1.12). The risk of HNC was associated with MetS in participants not performing regular exercise (HR: 1.05; 95% CI: 1.00–1.10). MetS was a statistically significant risk factor with a higher risk estimate of HNC regardless of BMI.

## 4. Discussion

The incidence of MetS continues to rise globally, particularly in developed countries in which diabetes and obesity rates are increasing [2,3]. MetS is associated with many diseases, and recent studies have demonstrated an increased risk for several cancers, including liver, colorectal, bladder, endometrial, pancreatic, and breast cancers [12,13,14,15]. Another previous research study established an association between MetS and thyroid cancer [16], and particularly with more aggressive forms of papillary thyroid cancer [16,23]. However, the relationship between MetS and HNCs other than thyroid cancer has not been thoroughly researched. To the best of our knowledge, this is the first large population-based study to examine the association of MetS with HNC and individual subtypes of cancers rather than examining the association of HNC with individual components of MetS.

In this study, the risk for developing HNC was significantly associated with MetS after adjusting for all confounders in Model ^3^ (HR = 1.06, 95% CI: 1.01–1.10). Notably, oral cavity cancer (HR = 1.12) and laryngeal cancer (HR = 1.18) were significantly associated with MetS. Interestingly, the association between MetS and a risk of cancer was not observed for other types of HNCs, including salivary gland cancer, nasopharyngeal cancer, oropharyngeal cancer, hypopharyngeal cancer, and sinonasal cancer. This suggests that the relationship between MetS and HNC may be specific to certain anatomical sites within the head and neck region.

Further analysis focusing on the individual components of MetS revealed that elevated fasting glucose (HR = 1.04) and elevated blood pressure (HR = 1.08) were significantly associated with an increased risk of HNC. In the subtype analysis, elevated fasting glucose was correlated with laryngeal cancer and elevated blood pressure with laryngeal, hypopharyngeal, and oral cancer. Hyperglycemia is a critical factor in the pathophysiology of MetS. Hyperglycemia, hyperinsulinemia, and insulin resistance increase proliferation and angiogenesis and destroy DNA molecules due to oxygen-active forms caused by excess glucose, cell migration, and apoptosis [24,25,26]. Previous studies have suggested that MetS elevates the risk of cancer through an induction of changes in insulin receptors and activation of growth and transcription factors [27,28]. Several studies have evaluated the influence of diabetes on HNC risk [29,30,31]. Although numerous studies have documented an association between hypertension and cancer, not all yielded conclusive results. In a meta-analysis of ten longitudinal studies conducted in 2002, individuals with hypertension were found to have a higher risk of overall cancer mortality with an odds ratio of 1.23 (95% CI: 1.11–1.36). A further analysis based on 13 case–control studies revealed a significant positive association between hypertension and renal cell carcinoma mortality with an odds ratio of 1.75 (95% CI: 1.61–1.90) [32]. Subsequent studies have corroborated these findings by confirming the association between hypertension and an increased incidence of renal cell carcinoma [33,34]. However, hypertension has not received significant attention as a risk factor for other types of cancer. Causal relationships between hypertension and cancers are difficult to establish; reverse causality and false evidence due to biases are difficult to rule out. Nonetheless, shared risk factors and mechanisms may contribute to these conditions. One hypothesis suggests that chronic inflammation, which is associated with an increased predisposition to cancer, may also be involved in the development of hypertension through vascular inflammation [35].

HNCs are closely related to tobacco use and alcohol consumption. Our study demonstrated that participants with MetS exhibited a higher risk estimate for the development of HNC even in non- or ex-smokers (HR: 1.09). This was also true for individuals with no or mild alcohol consumption (HR: 1.07). HNC incidence increases for current smokers and heavy drinkers, regardless of the presence of MetS. Conversely, for never or ex-smokers or infrequent alcohol consumers, the risk of HNC rises with the presence of MetS. This suggests that MetS acts as an independent risk factor for HNC, which is separate from the influence of smoking and alcohol consumption.

In previous studies, the association between obesity and the incidence of HNC remained controversial [6,7,8,9]. In most studies, HNC is associated with a lower BMI compared with a normal and higher BMI [36]. Gaudet et al. suggested that a low BMI was associated with a higher risk of HNC, regardless of smoking and drinking [7]. Furthermore, a large cohort study conducted at our institution, which analyzed the correlation between BMI and HNC, also found similar results to those found in previous studies [37].

This study highlights the importance of considering MetS as a potential risk factor for HNC and emphasizes the need for comprehensive management and prevention strategies targeting the components of MetS. Further research, including well-designed case–control studies, is warranted to gain a deeper understanding of the underlying mechanisms linking MetS to specific types of HNCs and to explore potential preventive and therapeutic interventions.

### Strengths and Limitations

Our study had several notable strengths. First, we had a substantial study population, consisting of approximately one-fourth of the adult Korean population, to provide an adequate representative sample. Moreover, the longitudinal design covered a ten-year follow-up period, enabling the collection of comprehensive data on cancer development. Second, the reliability of the data was enhanced by the accessibility of Korea’s healthcare system, which ensures universal coverage for all citizens. Third, the study’s rigorous methodology included only pathologist-diagnosed cancer cases and precise MetS diagnoses based on strict diagnostic criteria.

The study had certain limitations that should be acknowledged. First, most participants were of Korean descent, which limits the generalizability of the findings to other racial or ethnic groups. Second, the data did not include information on medications or other health conditions that could potentially increase the risk of cancer, limiting our ability to account for these factors. Third, the study did not consider the potential selection bias that could result from collecting data through self-reported questionnaires regarding prior medical history and health-related behaviors. Fourth, comprehensive data on HPV infection, which is associated with HNC, were not available for all patients. Additionally, industrial data were not available. Last, this study was conducted as a retrospective cohort study spanning a duration of 10 years; participants’ lifestyles, such as their smoking or drinking habits, may have undergone changes during this observational period. Capturing and incorporating these changes into the data analysis proved challenging. Also, exploring a causal relationship between changes in BMI, waist circumference, and the risk of HNC was not within the scope of this cohort study.

## 5. Conclusions

In this large cohort study, we found a significant association between MetS and the incidence of HNC. Participants with MetS had a higher risk of developing HNC, particularly oral cavity cancer and laryngeal cancer. Elevated fasting glucose and blood pressure present in MetS were independently associated with an increased HNC risk. Further research on this subject may lead to recommendations that patients diagnosed with MetS, especially diabetes and hypertension, should undergo regular screenings for HNC.

## Figures and Tables

**Figure 1 cancers-15-04118-f001:**
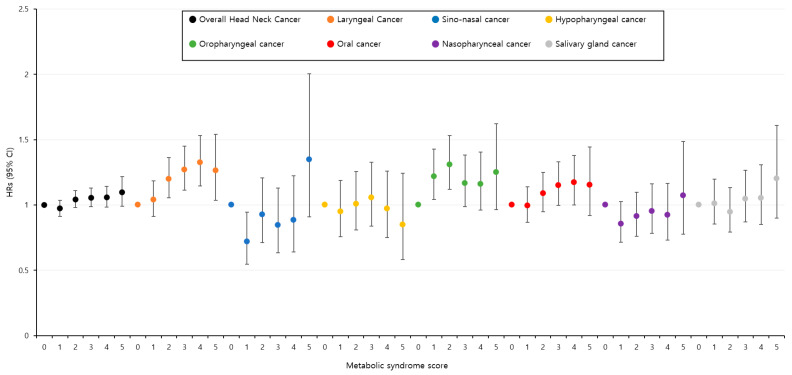
Hazard ratios for incidence of head and neck cancer and subtypes according to the score of metabolic syndrome components.

**Table 1 cancers-15-04118-t001:** General characteristics of the participants at baseline in 2009 and head and neck cancer identified 2010–2018 (n = 9,598,085).

	Overall Head and Neck Cancer, No. (%)		
Parameter	Yes (n = 10,732)	No (n = 9,587,353)	*p*-Value
Age (years)			<0.001
<40	744 (6.9)	3,029,454 (31.6)	
40–64	6547 (61)	5,336,521 (55.7)	
≥65	3441 (32.1)	1,221,378 (12.7)	
Gender			<0.001
Male	8500 (79.2)	5,212,301 (54.4)	
Female	2232 (20.8)	4,375,052 (45.6)	
Smoking status			<0.001
Non-smoker	4273 (39.8)	5,768,089 (60.2)	
Ex-smoker	2239 (20.9)	1,323,213 (13.8)	
Current smoker	4220 (39.3)	2,496,051 (26)	
Drinking status			<0.001
Non-drinker	4726 (44)	4,934,585 (51.5)	
Mild drinker	4441 (41.4)	3,888,132 (40.5)	
Heavy drinker	1565 (14.6)	764,636 (8)	
Regular exercise	2263 (21.1)	1,706,149 (17.8)	<0.001
Low income	2199 (20.5)	1,875,717 (19.6)	0.016
Diabetes	1728 (16.1)	828,911 (8.7)	<0.001
Hypertension	4618 (43)	2,462,226 (25.7)	<0.001
Dyslipidemia	2491 (23.2)	1,733,875 (18.1)	<0.001
Body mass index (kg/m^2^)	23.69	23.7	0.812
Waist circumference (cm)	83.16	80.2	<0.001
Systolic BP (mmHg)	127.13	122.42	<0.001
Diastolic BP (mmHg)	78.43	76.31	<0.001
Fasting glucose (mg/dL)	102.69	97.23	<0.001
HDL cholesterol (mg/dL)	55.49	56.5	0.002
Triglycerides (mg/dL)	126.8 (125.45–128.17)	112.66 (112.62–112.7)	<0.001
Metabolic syndrome score			<0.001
0	1617 (15.1)	2,604,250 (27.2)	
1	2519 (23.5)	2,592,085 (27)	
2	2684 (25)	1,999,486 (20.9)	
3	2107 (19.6)	1,362,223 (14.2)	
4	1317 (12.2)	769,221 (8)	
5	488 (4.6)	260,088 (2.7)	

BP, blood pressure; HDL, high density lipoprotein. Values are mean ± SE or % ± SE using Student’s *t*-test and the χ^2^ test.

**Table 2 cancers-15-04118-t002:** Multivariate Cox proportional hazards model for incidence of head and neck cancer and subtypes according to the presence of metabolic syndrome.

Cancer	Metabolic Syndrome	No. of Patients	Person- Years	Incidence Rates	Hazard Ratio (95% Confidence Interval)		
					Model ^1^	Model ^2^	Model ^3^
Head and neck cancer							
	Yes	3912	19,527,470	0.2	1.75 (1.68–1.82)	1.07 (1.03–1.11)	1.06 (1.01–1.10)
	No	6820	59,435,425	0.115	1 (reference)	1 (reference)	1 (reference)
Oral cavity cancer							
	Yes	825	19,536,667	0.018	1.80 (1.65–1.96)	1.13 (1.04–1.24)	1.12 (1.03–1.23)
	No	1400	59,451,603	0.023	1 (reference)	1 (reference)	1 (reference)
Salivary gland cancer							
	Yes	434	19,537,796	0.022	1.58 (1.40–1.77)	1.10 (0.97–1.24)	1.09 (0.96–1.23)
	No	839	59,453,425	0.014	1 (reference)	1 (reference)	1 (reference)
Nasopharyngeal cancer							
	Yes	357	19,538,140	0.018	1.46 (1.28–1.65)	1.05 (0.92–1.20)	1.05 (0.92–1.19)
	No	744	59,453,685	0.012	1 (reference)	1 (reference)	1 (reference)
Oropharyngeal cancer							
	Yes	615	19,537,376	0.031	1.56 (1.42–1.72)	0.98 (0.89–1.09)	0.97 (0.88–1.08)
	No	1199	59,452,193	0.02	1 (reference)	1 (reference)	1 (reference)
Hypopharyngeal cancer							
	Yes	348	19,538,399	0.017	1.82 (1.59–2.08)	1.02 (0.89–1.17)	1.02 (0.89–1.17)
	No	581	59,454,974	0.01	1 (reference)	1 (reference)	1 (reference)
Sinonasal cancer							
	Yes	194	19,538,730	0.01	1.71 (1.43–2.04)	1.08 (0.90–1.29)	1.06 (0.89–1.27)
	No	345	59,455,278	0.006	1 (reference)	1 (reference)	1 (reference)
Laryngeal cancer							
	Yes	1190	19,535,161	0.061	2.03 (1.89–2.19)	1.18 (1.09–1.27)	1.18 (1.09–1.27)
	No	1782	59,450,381	0.03	1 (reference)	1 (reference)	1 (reference)

Incidence rates per 1000 person-years. Model ^1^: unadjusted. Model ^2^: adjusted for age, gender, smoking, alcohol consumption, regular exercise, and income. Model ^3^: adjusted for age, gender, smoking, alcohol consumption, regular exercise, income, diabetes, and hypertension.

**Table 3 cancers-15-04118-t003:** Multivariate Cox proportional hazards model for incidence of head and neck cancer according to the presence or absence of metabolic syndrome components.

	No. of Patients	Person-Years	Incidence Rates	Hazard Ratio (95% Confidence Interval)		
				Model ^1^	Model ^2^	Model ^3^
High waist circumference						
Yes	2718	15,436,055	0.176	1.40 (1.34–1.46)	1.01 (0.97–1.06)	1.01 (0.97–1.06)
No	8014	63,526,839	0.126	1 (reference)	1 (reference)	1 (reference)
High fasting glucose						
Yes	4590	24,537,413	0.187	1.66 (1.60–1.72)	1.05 (1.01–1.09)	1.04 (1.00–1.08)
No	6142	54,425,482	0.113	1 (reference)	1 (reference)	1 (reference)
High blood pressure						
Yes	6568	34,137,709	0.192	2.07 (1.99–2.15)	1.09 (1.05–1.14)	1.08 (1.04–1.13)
No	4164	44,825,185	0.093	1 (reference)	1 (reference)	1 (reference)
High triglycerides						
Yes	4756	27,591,045	0.172	1.48 (1.43–1.54)	1.07 (1.03–1.11)	1.02 (0.99–1.06)
No	5976	51,371,849	0.116	1 (reference)	1 (reference)	1 (reference)
Low HDL cholesterol						
Yes	3284	21,603,446	0.152	1.17 (1.12–1.22)	1.02 (0.97–1.06)	1.03 (0.99–1.08)
No	7448	57,359,448	0.13	1 (reference)	1 (reference)	1 (reference)

HDL, high density lipoprotein. Incidence rate per 1000 person-years. Model ^1^: unadjusted. Model ^2^: adjusted for age, gender, smoking, alcohol consumption, regular exercise, and income. Model ^3^: adjusted for age, gender, smoking, alcohol consumption, regular exercise, income, diabetes, and hypertension.

**Table 4 cancers-15-04118-t004:** Analysis of factors potentially associated with head and neck cancer according to the presence or absence of metabolic syndrome.

Parameter	Metabolic Syndrome	Number	Event	Person-Years	Incidence Rates	Hazard Ratio (95% Confidence Interval)
Age (years)						
<40	Yes	342,900	116	2,836,339	0.041	1.13 (0.92–1.38)
	No	2,687,298	628	22,312,742	0.028	1 (reference)
40–64	Yes	1,465,717	2297	12,109,039	0.19	1.05 (0.10–1.10)
	No	3,877,351	4250	32,159,366	0.132	1 (reference)
≥65	Yes	586,827	1499	4,582,091	0.327	1.05 (0.98–1.12)
	No	637,992	1942	4,963,316	0.391	1 (reference)
Gender						
Male	Yes	1,393,195	3094	11,302,299	0.274	1.08 (1.03–1.13)
	No	3,827,606	5406	31,429,032	0.172	1 (reference)
Female	Yes	1,002,249	818	8,225,170	0.099	1.17 (1.06–1.28)
	No	3,375,035	1414	28,006,392	0.051	1 (reference)
Never- or ex-smoker						
Yes	Yes	1,770,124	2474	14,454,066	0.171	1.09 (1.04–1.15)
	No	5,327,690	4038	44,049,312	0.092	1 (reference)
No	Yes	625,320	1438	5,073,403	0.283	1.05 (0.99–1.12)
	No	1,874,951	2782	15,386,112	0.181	1 (reference)
Non- or mild drinker						
Yes	Yes	2,154,161	3309	17,562,691	0.188	1.07 (1.02–1.12)
	No	6,677,723	5858	55,128,638	0.106	1 (reference)
No	Yes	241,283	603	1,964,778	0.307	1.03 (0.93–1.14)
	No	524,918	962	4,306,786	0.223	1 (reference)
Regular exercise						
Yes	Yes	454,650	846	3,725,877	0.227	1.07 (0.98–1.17)
	No	1,253,762	1417	10,369,431	0.137	1 (reference)
No	Yes	1,940,794	3066	15,801,592	0.194	1.05 (1.00–1.10)
	No	5,948,879	5403	49,065,993	0.11	1 (reference)
Body mass index (kg/m^2^)						
<25	Yes	943,413	1905	7,612,933	0.25	1.13 (1.07–1.19)
	No	5,523,085	5342	45,536,885	0.117	1 (reference)
≥25	Yes	1,452,031	2007	11,914,537	0.17	1.09 (1.01–1.16)
	No	1,679,556	1478	13,898,539	0.106	1 (reference)

Incidence rates per 1000 person-years. Adjusted for age, gender, smoking status, alcohol consumption, regular exercise, income, diabetes, and hypertension.

## Data Availability

Data are only available upon request due to data sharing restrictions. The data presented in this study are available upon request from the corresponding author.

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
