# Peer review of "Association of Metabolic Syndrome with the Risk of Head and Neck Cancer: A 10-Year Follow-Up Study of 10 Million Initially Healthy Individuals"

_cancers, 2023, doi:10.3390/cancers15164118_

Round 1

Reviewer 1 Report

This study examined the relationship between Metabolic Syndrome and Head and Neck Cancer incidence rate in a large sample of Korean adults using data from medical insurance records.  The authors found significant positive associations between MetS and various type of HNC incidences. In my opinion, the very large sample size makes the study unique and important. However, there are several details in the methodology that needs clarification and the way how the results are presented could be improved.

My specific comments are the follows:

Abstract:

1.      Why do the authors say that it is a retrospective study (line 12)? Seems to me like a prospective cohort design.

2.      Line 16: please add 95% CI when mentioning the HNC risk in those with MetS.

Introduction

3.      Line 42: Why is it a HNC meta analysis? According to the classification given earlier in the text, thyroid cancers do not belong to the HNC group.

4.      The overall justification of the analysis could be made stronger. For example, could the authors give some information on the importance if HNC in Korea? (Ie. prevalence, trends, etc?)

Materials and Methods

5.      The population of Korea is around 50million. The authors say that the insurance data used for this analysis covers 97% of the population, but they present the analysis on 10million individuals - i.e. those who went through medical checkup in 2009. How are people selected for the check up in each year? Most importantly, the authors will need to explain whether this sample is representative to the entire Korean population? Or how it differs from it (potentially can be elaborated on in the discussion section when discussing limitations).

6.      Line 68: Authors say that individuals with missing data was excluded. But there is no information on what data was considered for this. Any individuals on missing data on variables presented in table 1 were excluded? Or only those variables that were included in the regression models?

7.      Data on medical history and health behaviour was also collected during the 2009 health check-up? In table 1, there is information on Income as well. Does this come from this self-reported questionnaire too? If yes, how was this asked? In the medical examination, how is Blood pressure measured? Is this procedure standardised across the country?

8.      In terms of statistical analysis, I`m wondering how the authors dealt with losses of follow up. For example, participants who died of other reasons during the 10-year period, or those whose mortality data was not available (i.e. moved abroad)?

9.      In the Statistical analysis section, it would be good to describe what adjustments were made (i.e. what are the 3 models) and please give some rationale for the variables adjusted for.

Results

10.   In General, table 1 could be formatted better because in the current form it is not easy to see which categories belong to the same variable

11.   Table 1: In a footnote, please indicate what statistical tests were used to calculate p-values.

12.   In table 1, there is information on existing diabetes. How was this information taken into account in the analysis?

13.   Table 2: Incidence rate per 1000 individuals or per 1000 person-years? - please clarify

14.   Table 2: Why is low income not adjusted for? It seems to be a good indicator of Socio-economic Position.

15.   Figure 1 - please add label to the horizontal axis. The second line of the figure title is the description of the result, not the title. Do these HRs indicate adjusted or unadjusted associations. If adjusted, please say what they were adjusted for.

16.   Table 3: Please explain better what "number" is in the heading of the table.

17.   In the subgroup analysis, it would be better to look at never smokers independently, rather than combine them with ex-smokers, as the risk for ex-smokers might be still considerable and unclear when they stopped smoking

Discussion

18.   In terms of generalisability, please consider the representativeness of the analytical sample to the entire Korean population.

19.   In terms of information bias, the authors should also consider that most covariates were measured with self-report.

References

20.   The formatting of some of the references needs to be checked.

Reviewer 2 Report

This is an interesting study that takes excellent advantage of a large epidemiologic cohort using data from almost 10 million persons enrolled in the Korean insurance system.

My comments are as follows:

Abstract. The abstract states as the rationale for the study that research on the association between metabolic syndrome and the incidence of head and neck cancer “has rarely been conducted”.  This is a week argument for why this topic needs to be studied.  More compelling reasons if true might be that prior research on the topic needs confirmation in a larger study or that research in animal models or mechanistic studies suggest there might be an association that should be better understood.

Abstract. A better way to expression the sentence in lines 21-23 might beHNC was associated with MetS “even among never or ex-smokers… or among none or mild drinkers”.

Introduction line 44.  The statement “Early diagnosis of MetS is necessary in individuals with malignant tumors.” The sentence seems out of place, and it is not clear why early diagnosis of MetS is necessary in people who already have cancer.  The introductory paragraph is missing any evidence from prior studies that obesity or MetS is associated with head and neck cancer. This information should be included.

Section 2.3  Please make it clear if the MetX information was from the baseline health checkup (and not at for example, other medical record data in 2010-2018).

Table 1.  A more precise table title would be helpful; for example – General characteristics of the study participants at baseline in 2009 and head and neck cancer identified 2020-2018.

Table 2 and Table S1.  Use a number or symbol as a superscript (e.g., 1 after “model 1” and against the footnote below the table. Initially I thought the authors had not provided an explanation of what was in the models.

Table 2. I think the table doesn’t include the data for laryngeal cancer. The text refers to largyneal as having a strong association with MetS, but the authors cite the HR for sinonasal cancer not laryngeal cancer. 

Line 125. The word “Strong” is used to describe the findings when apparently what is being discussed is statistical significance, which is not the same thing.

Line 127 and Figure 1.  The term “prevalence” is used, when I think the authors are writing about incidence. Please clarify throughout. Line 127-128 – note whether there is significant linear trend for each of the subtypes

Figure 1 title and Supplement table S.1 and S.2 add to title somewhere the words “by subtype” to make it clearer that you are analyzing by cancer subtype.

Section 3.3.  There are a very large number of comparisons being done; can some correction or some way to account for the large number of tests??  The paragraph is so detailed that I can’t make any sense of it.

Section 3.4 Again the authors state “MetS was a strong risk factor” when it is not clear if they are referring to the statistical test or the magnitude of the hazard ratio.

Discussion. Although lines 180 and 181 state that there have been no studies of MetS and head and neck cancer and subtypes, there certainly are studies of obesity and head and neck cancer and possibly other MetS subcomponents that should be mentioned here.
